# Effects of a Prenatal Lifestyle Intervention in Routine Care on Maternal Health Behaviour in the First Year Postpartum—Secondary Findings of the Cluster-Randomised GeliS Trial

**DOI:** 10.3390/nu13041310

**Published:** 2021-04-15

**Authors:** Kristina Geyer, Monika Spies, Julia Günther, Julia Hoffmann, Roxana Raab, Dorothy Meyer, Kathrin Rauh, Hans Hauner

**Affiliations:** 1Else Kröner-Fresenius-Centre for Nutritional Medicine, Institute of Nutritional Medicine, School of Medicine, Technical University of Munich, Georg-Brauchle-Ring 62, 80992 Munich, Germany; k.geyer@tum.de (K.G.); monika.spies@tum.de (M.S.); julia.guenther@tum.de (J.G.); julia.hoffmann@tum.de (J.H.); roxana.raab@tum.de (R.R.); dora.meyer@tum.de (D.M.); kathrin.rauh@tum.de (K.R.); 2Competence Centre for Nutrition (KErn), Am Gereuth 4, 85354 Freising, Germany

**Keywords:** postnatal, postpartum lifestyle, maternal diet, dietary behaviour, physical activity, smoking, obesity prevention, lifestyle intervention, primary care

## Abstract

Lifestyle interventions during pregnancy were shown to beneficially influence maternal dietary behaviour and physical activity, but their effect on health behaviour after delivery is unclear. The objective of this secondary analysis was to investigate the sustained effect of a lifestyle intervention in routine care on maternal health behaviour during the first year postpartum. The cluster-randomised controlled “Healthy living in pregnancy” (GeliS) study included 2286 pregnant women. Data on maternal health behaviour were collected at 6–8 weeks (T1pp) and one year postpartum (T2pp) using validated questionnaires. The intervention group showed a lower mean intake of fast food (T1pp: *p* = 0.016; T2pp: *p* < 0.001) and soft drinks (T1pp: *p* < 0.001), a higher mean intake of vegetables (T2pp: *p* = 0.015) and was more likely to use healthy oils for meal preparation than the control group. Dietary quality rated by a healthy eating index was higher in the intervention group (T1pp: *p* = 0.093; T2pp: *p* = 0.043). There were minor trends towards an intervention effect on physical activity behaviour. The proportion of smokers was lower in the intervention group (*p* < 0.001, both time points). The lifestyle intervention within routine care modestly improved maternal postpartum dietary and smoking behaviours.

## 1. Introduction

Healthy lifestyle habits during the time of pregnancy and postpartum are important for optimising maternal and child health in the short- and long-term [1,2]. The modern Western lifestyle, with a high-caloric diet, low physical activity (PA) and high risk of weight gain may be detrimental for pregnancy and the early postpartum period [2,3]. In Germany, 67% of pregnant women gain weight in excess of the recommendations set forth by the Institute of Medicine (IOM) [4,5], increasing the likelihood of postpartum weight retention [6,7,8], weight gain in subsequent pregnancies [9] and overweight later in life [10]. Growing evidence suggests that a healthy lifestyle during pregnancy, but also after delivery, is associated with multiple benefits such as with an adequate body mass index (BMI) [11,12,13], cardio-metabolic benefits [14,15,16], aerobic fitness, insulin sensitivity and improvements in overall psychological wellbeing [15,17,18,19]. In order to support women during pregnancy and the postpartum period, recommendations on optimal dietary behaviour and regular PA have been published [20,21,22,23].

There is evidence that antenatal lifestyle interventions are able to improve maternal diet and PA during gestation [24,25] and achieve a reduction in excessive gestational weight gain (GWG) [26]. However, adhering to a healthy lifestyle during the postpartum period has been shown to be challenging, and several studies indicate a decrease in dietary quality and PA after delivery [27,28,29,30,31]. This might be attributed to factors, such as lack of time and sleep, parenting duties and prioritising competing responsibilities over health, which prevent women from implementing or maintaining a healthy diet and PA level [32]. Therefore, supporting women in achieving a healthy lifestyle already in pregnancy might help to sustain healthy routines after delivery. While the effects of lifestyle interventions on maternal diet and PA during pregnancy have been extensively studied [24,25,26], lasting intervention effects on maternal health behaviour in the postpartum period are poorly explored [33,34,35,36]. Moreover, interventions were primarily conducted under fairly controlled conditions in academic settings [26,37] rather than in a primary care setting. To address this research gap, this secondary analysis of the large-scale, cluster-randomised “Gesund leben in der Schwangerschaft” (GeliS; “Healthy living in pregnancy”) study investigated if a comprehensive lifestyle intervention, embedded in routine care, was able to influence maternal lifestyle during the first year postpartum.

## 2. Materials and Methods

### 2.1. The GeliS Study: Design and Setting

The GeliS study is a prospective, multicentre, cluster-randomised, controlled, open intervention trial that took place in five administrative regions in Bavaria, Germany. One control and one intervention district per region were matched according to birth figures, sociodemographic and geographic criteria via a pairwise cluster-randomisation. The study was conducted in participating gynaecological and midwifery practices within the 10 districts. More information on the study design and setting is available in the published study protocol [38]. In total, 2286 pregnant women were recruited in the GeliS trial [39]. The primary goal of the GeliS study was to reduce the proportion of women with excessive GWG according to the IOM criteria [5] through a comprehensive lifestyle intervention programme alongside routine care visits. Results on primary and secondary endpoints have already been published [8,39,40,41,42,43,44,45]. This secondary analysis focuses on the effect of the GeliS lifestyle intervention programme on maternal health behaviour during the first 12 months postpartum. For this purpose, we investigated differences in health-related lifestyle behaviours, including diet, PA and smoking, between the intervention (IG) and control groups (CG). Additionally, we examined the intervention effect among several subgroups. The study complied with local regulatory requirements and laws and the declaration of Helsinki. The Technical University of Munich Ethics Committee approved the study protocol. The study was registered in the ClinicalTrials.gov Protocol Registration System (NCT01958307). 

### 2.2. Participants

Recruitment was carried out by medical personnel between 2013 and 2015 at 71 gynaecological and midwifery practices. Pregnant women who met the following inclusion criteria were considered for the study: Pre-pregnancy BMI ≥ 18.5 kg/m^2^ and ≤ 40.0 kg/m^2^, singleton pregnancy, age between 18 and 43 years, sufficient German language skills, stage of pregnancy before the 12th week of gestation and provision of written informed consent. Severe complications that interfered with the intervention were reasons for exclusion [38]. During the follow-up period, women were considered to be drop-outs if they could no longer be reached, did not offer contact information or declined further participation [8].

### 2.3. Lifestyle Intervention

The lifestyle intervention programme comprised four comprehensive face-to-face counselling sessions (12th–16th, 16th–20th and 30th–34th week of gestation, and 6–8 weeks postpartum) alongside routine care delivered by trained midwives, medical personnel or gynaecologists. The counselling focused on an adequate GWG as recommended by the IOM [5]. Women were encouraged to maintain healthy dietary and PA behaviours according to national and international guidelines [20,23,46]. They were informed about the principles of a healthy diet, were motivated to engage in at least 150 min of moderate-intensity activity per week and encouraged to increase their daily activity level. Additionally, women were educated about the importance of breastfeeding and the avoidance of smoking and alcohol intake during pregnancy and lactation. Women in the CG received routine antenatal care, along with a leaflet that provided general recommendations on a healthy lifestyle and breastfeeding. Details on the GeliS lifestyle intervention and counselling content have been published elsewhere [38].

### 2.4. Data Collection and Processing

Data on baseline maternal characteristics were obtained from a screening questionnaire before the 12th week of gestation. Maternal dietary, PA and smoking data were collected via questionnaires in early pregnancy (before the 12th week of gestation, used as baseline) and late pregnancy (after the 29th week of gestation), and at two time points during the first year postpartum (T1pp: 6–8 weeks postpartum; T2pp: One year postpartum). The intervention effect on antenatal dietary and PA behaviours has already been published elsewhere [41,42]. The analyses presented herein focus on the dietary and PA behaviours during the postpartum period as well as smoking behaviour, for which all four time points of data collection were considered.

Dietary variables were collected via a self-administered and validated food frequency questionnaire (FFQ) originally used for the “German Health Interview and Examination Survey for Adults” (DEGS) trial conducted by the Robert Koch Institute, Berlin, Germany [47]. The utilised DEGS-FFQ was slightly altered from the original version and consisted of 54 questions relating to consumption frequency and portion size of food items and dietary behaviour over the previous four weeks. Additionally, four questions targeted food preparation and dietary choices, vegetarianism and the frequency of fresh food preparation. For the calculation of mean daily intakes, food items were grouped into 17 food groups as recommended by the developers of the DEGS-FFQ (personal communication: Dr. G. Mensink, Robert Koch Institute, 2018). In case of over-reporting of food intake or missing amounts of more than 20 out of 54 food items, questionnaires were excluded from the analysis as described previously [42]. Energy, macronutrient and fibre intake were estimated using the German food composition database (“Bundeslebensmittelschlüssel”, version 3.02) using the OptiDiet PLUS software (version 6.0, GOE mbH, Linden, Germany). Women under- or over-reporting energy intake were precluded from the statistical analysis of energy and macronutrient intake as described previously [42]. The DEGS-Healthy Eating Index (DEGS-HEI) developed by the Robert Koch Institute [48] was calculated to evaluate the dietary quality based on the DEGS-FFQ. The index has a range from 0 to 100, with values closer to 100 indicating better adherence to the German Nutrition Society recommendations [48]. 

A slightly modified version of the validated Pregnancy Physical Activity Questionnaire (PPAQ) [49] was used to collect data on duration, frequency and intensity of PA behaviour. Participants reported the time spent in 32 activities over the past four weeks and had the option to name two additional sport activities that were not listed in the PPAQ in two open-ended questions. According to the calculation instructions of the PPAQ [50] respecting the 2011 Compendium of Physical Activity [51], the estimated amount of time spent in each activity per week was calculated and expressed as multiples of resting energy expenditures in metabolic equivalent of task (MET)-hours per week. Thereby, PA could be classified in different categories of total activity, activity types and activity intensities. The over-reporting of PA data was defined according to others [25] and has been previously reported [41]. The threshold of ≥7.5 MET-h/week in sport activities of moderate intensity or greater was used to indicate whether women achieved the activity level set in national and international PA recommendations [23,52]. This procedure was recommended by the developers of the PPAQ (personal communication: Prof. L. Chasan-Taber, University of Massachusetts Amherst, 2018) and has been applied previously [41].

Smoking behaviour of women was assessed with the question ‘Do you currently smoke?’, which was part of the questionnaires.

### 2.5. Statistical Analysis

The power calculation was based on excessive GWG as the primary endpoint and has already been described elsewhere [38]. Women were included in the present analysis if they provided at least either a diet or a PA questionnaire at T1pp or T2pp and were not pregnant at T2pp. These women were likewise considered for the analysis of smoking behaviour. Group differences in diet and PA variables were estimated using linear regression models fit with generalised estimating equations (GEE), as recommended for cluster-randomised trials [53]. The models were adjusted for pre-pregnancy BMI category, age, parity, baseline dietary assessment for dietary variables or baseline PA assessment for PA variables, respectively, and time interval between questionnaire completion date and offspring birth date. Dichotomised variables were compared using binary logistic regression models fit with GEEs and adjusted for the same covariates. The changes in dietary and PA behaviours between T1pp and T2pp (time effects) were investigated using linear mixed models for repeated measures adjusted for pre-pregnancy BMI category, age and parity. The group differences in smoking behaviour were analysed using binary logistic regression models fit with GEEs as described above. For the smoking assessment in early pregnancy, the regression model was adjusted for pre-pregnancy BMI category, age and parity. For late pregnancy, the model was additionally adjusted for baseline smoking assessment. Similar GEE models, as mentioned above, were applied in exploratory subgroup analyses according to maternal age, pre-pregnancy BMI category, educational level and parity. A post-hoc analysis on potential interactions of group assignment with these factors was performed to screen for influencing factors on the treatment effect. Across all analyses, a *p*-value < 0.05 was considered statistically significant. Data analysis was performed using SPSS software (IBM SPSS Statistics for Windows, version 26.0, IBM Corp, Armonk, NY, USA).

## 3. Results

### 3.1. Flow-Chart and Maternal Characteristics

In total, 2261 participants were recruited and assigned to the IG (*n* = 1139) or the CG (*n* = 1122) (Figure 1). Diet and/or PA data of 1899 women (84.0%) were collected on either one or both postpartum assessments (IG: *n* = 952; CG: *n* = 947). At 6–8 weeks postpartum, a total of 1791 women provided valid dietary data (IG: *n* = 898; CG: *n* = 893), 1812 provided valid PA data (IG: *n* = 907; CG: *n* = 905) and 1815 women provided data on smoking behaviour (IG: *n* = 905; CG: *n* = 910). During the first year postpartum, 101 women in the IG and 114 women in the CG were lost to follow-up, and 75 women were excluded from analyses due to subsequent pregnancies. At the end of the first year postpartum, valid dietary data from 1568 women (IG: *n* = 791; CG: *n* = 777), valid PA data from 1551 women (IG: *n* = 784; CG: *n* = 767) and smoking data from 1557 women (IG: *n* = 785; CG: *n* = 772) were available. With regard to all women who remained in the study until T1pp, the drop-out rate over the course of the one-year follow-up was 10.8%.

As shown in Table 1, mean age, self-reported pre-pregnancy weight, pre-pregnancy BMI and GWG were comparable between the IG and the CG. The majority of women (65.5%) had a normal pre-pregnancy weight, while 22.9% had overweight and 11.6% had obesity. The proportion of women with excessive GWG was similar in both groups (IG: *n* = 44.6%; CG: *n* = 44.9%). More women were primiparous at the time of inclusion in the IG (63.3%) compared to women in the CG (53.7%).

### 3.2. Postpartum Dietary Behaviour 

#### 3.2.1. Intake of Selected Food Groups

The mean daily intake for a variety of food groups is shown in Table 2. There was no significant evidence of an intervention effect on the mean intake of caffeinated beverages, fruits, nuts, cheese, meat and meat products, and sweets and snacks at both postpartum time points. At T1pp, women in the IG consumed significantly fewer soft drinks (*p* < 0.001), more dairy products (*p* = 0.012) and more fish (*p* < 0.001) than women in the CG. These between-group differences were no longer significant one year after delivery. At T2pp, women in the IG showed a significantly higher mean daily intake of vegetables compared to women in the CG (*p* = 0.015). Women in the IG consumed slightly less fast food (T1pp: *p* = 0.016; T2pp: *p* < 0.001). The mean consumption of caffeinated beverages increased significantly from 6–8 weeks postpartum to one year postpartum in both groups (*p* < 0.001 respectively), whereas the consumption of sweets and snacks decreased in both groups (*p* < 0.001 respectively).

#### 3.2.2. Food Preparation and Dietary Choices

Specific food preparation and dietary choices in the IG and CG are summarised in Table 3. It was not observed at T1pp, but one year after delivery, that a higher proportion of women in the IG chose whole grain bread compared to the CG. Women in the IG were more likely to use rapeseed oil and olive oil over other oils to prepare meat and fish throughout the postpartum period (T1pp: *p* = 0.004; T2pp: *p* = 0.011) and to prepare vegetables at T1pp (*p* = 0.012). The number of participants who prepared meals from fresh food at least five times a week increased in both groups from 6–8 weeks to one year postpartum (*p* < 0.001 in both groups). The proportion of vegetarians was comparable between groups.

#### 3.2.3. Energy and Macronutrient Intake

There was no significant evidence of differences between the groups in terms of mean energy and macronutrient intake (Table 4). Mean energy intake was 2236.99 kcal/day in the IG and 2227.84 kcal/day in the CG at T1pp. From T1pp to T2pp, the mean energy intake significantly decreased in both groups (*p* < 0.001 respectively). Alcohol consumption increased in both groups from early postpartum to one year postpartum; however, women in the IG consumed less alcohol than women in the CG one year after delivery (*p* = 0.014). The dietary quality, assessed by the HEI, was higher by trend in the IG at 6–8 weeks postpartum (*p* = 0.093). At one year postpartum, the HEI was significantly higher in the IG than in the CG (*p* = 0.043). Additional analyses revealed significant differences between the IG and CG in the following subgroups at T2pp: Women aged 26–35 years (*p* = 0.013), women with intermediate secondary education (*p* < 0.001) and multiparous women (*p* = 0.005) (Appendix A). Overall, dietary quality slightly increased in both groups over time (Table 4).

### 3.3. Postpartum Physical Activity Behaviour

Maternal postpartum PA behaviour is presented in Table 5. The level of total PA was slightly higher in the CG at T1pp compared to the IG (*p* = 0.023). Subgroup analyses (Appendix A) revealed a lower total PA in the IG compared to the CG in the subgroup of women aged 18–25 years (*p* = 0.044), the subgroup of women with normal weight (*p* = 0.047) and the subgroup of primiparous women (*p* = 0.001) (Appendix A). Over the course of the postpartum period, there was a significant increase in total PA in the IG (*p* < 0.001), but not in the CG (*p* = 0.061), resulting in a similar mean PA level in both groups one year after delivery (Table 5). Women in the IG showed a significantly higher level of occupational activity compared to the CG at T1pp (*p* = 0.016). No between-group differences in intensity of PA or other activity categories, including sport activity, were observed. In both groups, the mean MET-h/week in sedentary activities and the level of inactivity decreased significantly during the postpartum period (*p* < 0.001 respectively). In most other PA categories, levels increased significantly within the first year after delivery in both groups. The between-group differences in meeting the PA recommendations at T1pp (IG: 52.7%, CG: 46.7%, *p* = 0.060) and at T2pp (IG: 58.1%, CG: 55.2%, *p* = 0.957) were not statistically significant.

### 3.4. Maternal Smoking Behaviour

Figure 2 depicts the smoking behaviour in the IG and the CG from early pregnancy until one year postpartum. At baseline, smoking rate was 5.0% in each group. While the proportion of smokers in the CG did not change over the course of pregnancy, it decreased in the IG, so that significantly fewer women in the IG smoked in late pregnancy (IG: 3.8% vs. CG: 5.1%, *p* < 0.001). After delivery, the proportion of smokers increased in both groups. Nonetheless, at 6–8 weeks as well as at one year postpartum, the smoking rate was significantly lower in the IG compared to the CG (T1pp: 7.1% vs. 9.7%, *p* < 0.001; T2pp: 13.1% vs. 14.1%, *p* < 0.001). At both postpartum time points, the number of current smokers in the IG was consistently lower in the subgroup of women with normal BMI (T1pp: *p* = 0.019; T2pp: *p* = 0.017) and of women who went to intermediate secondary school (T1pp: *p* < 0.001; T2pp: *p* < 0.001) (Appendix A). Post-hoc interaction analysis revealed that educational level influenced the relationship between the intervention and smoking behaviour one year after delivery (*p* = 0.042).

## 4. Discussion

The aim of this secondary analysis was to examine whether the GeliS lifestyle intervention had a lasting effect on maternal health behaviour during the first year postpartum. The findings of the present analysis indicate that the GeliS intervention successfully improved some aspects of maternal postpartum health behaviour.

Women who had received lifestyle counselling showed a lower mean intake of fast food and soft drinks and consumed more vegetables than women in the CG. Furthermore, women in the IG were more likely to choose healthy oils for food preparation, and the overall quality of their diet assessed by the HEI was higher. In contrast, no difference was detected for total energy intake. This is in line with the observed intervention effect on women’s antenatal dietary behaviour [42] and suggests a sustained effect in the postpartum period. Since some differences in dietary behaviour were small, it is questionable if their clinical relevance is meaningful. Irrespective of group allocation, the overall dietary quality and vegetable intake increased and the consumption of sweets and snacks decreased during the postpartum phase. This corresponds to observations from Martin et al. [29], who also reported that dietary quality improved in the late vs. the early postpartum phase. Possible reasons for this might be the unique challenges present during the early postpartum period, such as fragmented sleep [54], fatigue [28] and prioritising the needs of the infant [27]. Due to childcare responsibilities and limited time after birth, healthy eating might not be a priority [28] and women might return to previous habits, disregarding healthy lifestyle advice [27].

Although some improvements in PA were evident during pregnancy [41], only minor between-group differences were found for the postpartum period. For instance, while a significantly higher proportion of women in the IG met the PA recommendations in late pregnancy [41], this significance disappeared in the postpartum period. A similar change was observed by Sanda et al. [55]. Nevertheless, only women in the IG increased total PA significantly over the course of the first postpartum year. Despite several minor trends towards a beneficial intervention effect, our data suggests that the GeliS lifestyle intervention was not successful in comprehensively influencing the PA behaviour beyond delivery. Irrespective of group allocation, the observed increase in TALIA over the postpartum period corresponds with previous observational studies in pregnant women [31,56]. Further, the proportion of women in the IG and the CG who met the PA recommendations in the postpartum period (T1pp: 52.7% vs. 46.7%; T2pp: 58.1% vs. 55.2%) exceeded the proportion of women meeting PA guidelines in the general German population (42.6%) measured by means of a written form of the European Health Interview Survey—Physical Activity Questionnaire [57]. The discrepancy with our data might be partly explained by the difficulty in estimating PA levels with the PPAQ. In both groups, the largest increase in PA levels from pregnancy to postpartum was observed for household activities, equally reported by Dodd et al. [25], and might be attributed to increasing caregiving tasks, which are not separately assessed by the PPAQ. 

We further observed a lower proportion of women who smoked in the IG compared to the CG during late pregnancy and the postpartum period. This is in contrast to the findings from a systematic review and meta-analysis [58]. Chamberlain et al. concluded that there is uncertainty if lifestyle interventions focusing not only on smoking, but also on enhancing maternal health overall, increase the chances of smoking cessation [58]. The smoking rate in our cohort was lower compared to data from a German cohort on smoking rates during pregnancy (10.9%) [59], and the general female adult population (20.8%) [60]. This discrepancy might be explained by a reporting bias or the data collection mode via questionnaires. Nevertheless, our subgroup analysis corresponds with these German-wide data indicating that smoking is less prevalent among adults with a higher educational level and most common in younger age groups [60]. In conclusion, the GeliS lifestyle intervention led to moderate beneficial effects on maternal health behaviours during pregnancy [41,42], which were partly sustained in the postpartum period.

Results from previous lifestyle intervention studies are inconclusive, and few randomised controlled trials (RCT) have investigated the sustained effects of mixed lifestyle interventions on maternal lifestyle behaviours up to 12 months postpartum. In contrast to our data, which include women with different BMI categories, two comparable large-scale RCTs recruited only women with overweight and obesity [25,33,34]. In the LIMIT study, many of the improvements in dietary quality and PA achieved during pregnancy were not sustained at four months postpartum [25]. In the UPBEAT trial, the positive changes in dietary behaviour observed in pregnant women with obesity persisted at six months [34] and three years postpartum [33]. However, no lasting effect on PA was achieved [33,34]. These results are, to some extent, in line with our observations. The Danish LiP-study found no effects on self-reported PA and eating habits among women with obesity at six months postpartum after receiving exercise classes during pregnancy [61]. In contrast, the ROLO trial reported continued compliance on dietary behaviours at three months postpartum [35]. In the aforementioned studies, follow-up assessment time points ranged from three months to three years [25,33,34,35,61], complicating the ability to determine whether intervention effects ultimately persist in the long-term. Altogether, findings from relevant studies depict a heterogeneous picture with respect to study design, population characteristics and type of intervention, which make direct comparisons to our data challenging. In contrast to the above-mentioned studies, our study was the only one that was implemented in a primary care setting. 

Several limitations of our analysis should be considered. Delivering the lifestyle counselling within the primary care system implied that the intervention was carried out by trained medical staff rather than lifestyle experts, e.g., dieticians or physiotherapists. Furthermore, our analyses largely relied on self-reported data from questionnaires. Variations in the timing of completion of the questionnaires is a consideration, and was controlled for by including the completion date as a confounding variable. Moreover, self-reported data are susceptible to inaccuracies, despite our efforts to reduce over- or underreporting. However, the applied DEGS-FFQ is a validated tool and especially useful for comparing dietary intake between groups [47]. While the PPAQ is a validated tool for the assessment of PA during pregnancy [49], it has not been formally validated for the postpartum period. Nevertheless, using the same questionnaires pre- and postnatally enabled us to compare results at different time points. 

Apart from these limitations, our study has several strengths that are worth mentioning. Importantly, the GeliS study was conceived and implemented as a public health RCT imbedded in routine care. The unique study design allowed the recruitment of a large cohort, comprising initially 2286 women, as the intervention was able to be delivered during pre- and post-natal visits. For this follow-up analysis, we included data from 1899 women, representing 83.7% of the original cohort. Compared to other studies [34,62], this is an exceptionally high retention rate. To the best of our knowledge, GeliS is the first large-scale trial that was conducted in a routine care setting that showed a detailed analysis of health behaviours of women in different BMI categories during the first year postpartum.

This analysis emphasises the need for future research on maternal health after delivery. In view of the challenge of postpartum weight retention [6,9], understanding key determinants of maintaining and even increasing healthy lifestyle habits during the postpartum period is a crucial public health issue. Because women face multiple barriers to adopting a healthy postpartum lifestyle [32], interventions should consider practical concerns. A technology-based approach may be able to alleviate barriers and provide tailored support within a woman’s daily life [63].

## 5. Conclusions

The GeliS trial is the only study conducted in a routine care setting presenting comprehensive information on maternal health behaviour up to 12 months postpartum. While previous results of the GeliS trial indicated that the lifestyle intervention was able to improve dietary and PA behaviour during pregnancy [41,42], this secondary analysis still showed slightly positive intervention effects on maternal health behaviour beyond the intervention phase. More attention should be given to a healthy postnatal lifestyle to mitigate the long-term obesity risk and related co-morbidities. More high-quality studies are needed to clarify the remaining uncertainty regarding the optimal approach to support a healthy postpartum lifestyle. Data on GeliS mother–child pairs up to the children’s fifth birthday will offer the opportunity to investigate whether differences in maternal postpartum lifestyle also have an impact on the children’s health behaviours and health-related outcomes.

## Figures and Tables

**Figure 1 nutrients-13-01310-f001:**
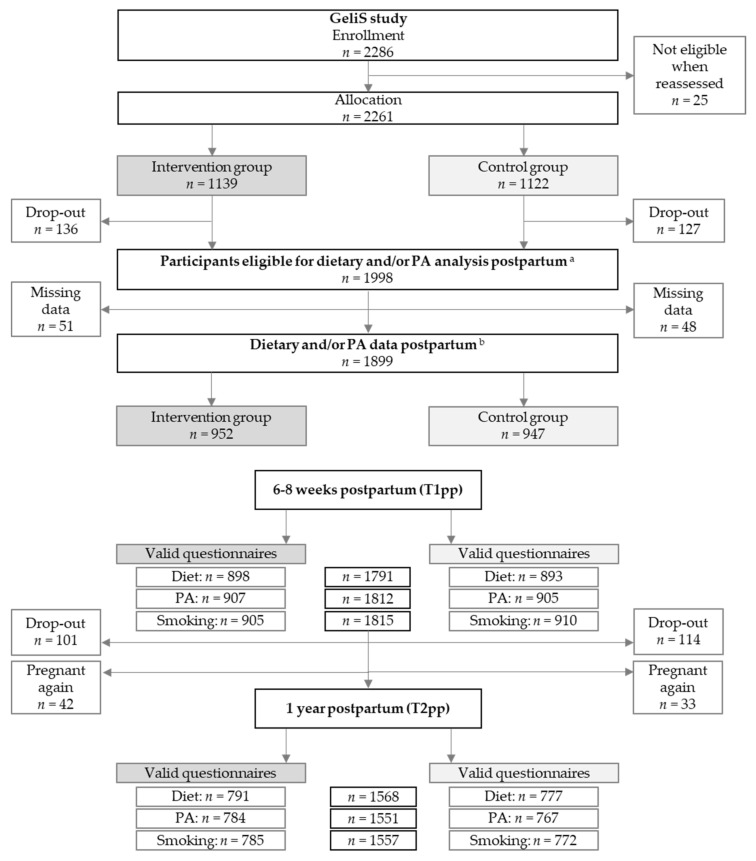
Flow of study participants. Abbreviations: T1pp: Assessment 6–8 weeks after delivery; T2pp: Assessment one year after delivery; PA: Physical activity. ^a^ Women who remained in the study until T1pp. ^b^ Women who provided PA and/or dietary data at T1pp or T2pp.

**Figure 2 nutrients-13-01310-f002:**
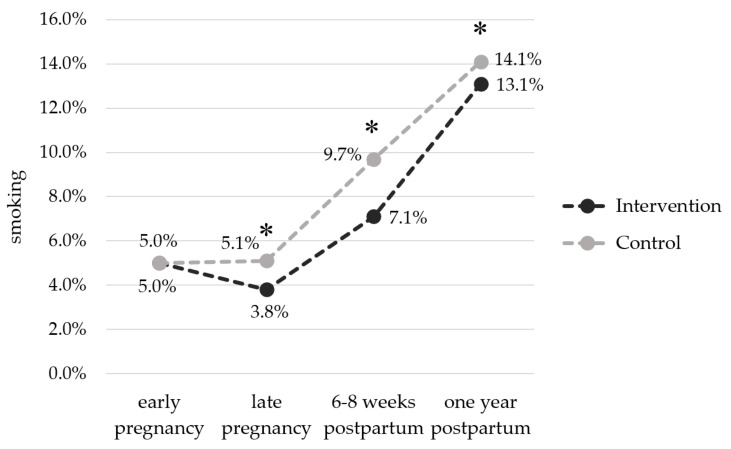
Percentage of smokers in the intervention and control groups from pregnancy to one year postpartum. Early pregnancy: IG: *n* = 47/931, CG: *n* = 46/922; late pregnancy: IG: *n* = 35/916, CG: *n* = 46/906; 6–8 weeks postpartum: IG: *n* = 64/905, CG: *n* = 88/910; one year postpartum: IG: *n* = 103/785, CG: *n* = 109/772. * *p* < 0.001.

**Table 1 nutrients-13-01310-t001:** Maternal characteristics.

	Intervention Group (*n* = 952)	Control Group (*n* = 947)	Total (*n* = 1899)
Pre-pregnancy age, years ^a^	30.3 ± 4.2	30.6 ± 4.5	30.4 ± 4.4
Pre-pregnancy weight, kg ^a^	68.3 ± 12.9	67.9 ± 13.5	68.1 ± 13.2
Pre-pregnancy BMI, kg/m^2 a^	24.4 ± 4.3	24.3 ± 4.5	24.3 ± 4.4
**Pre-pregnancy BMI category, *n* (%)**			
BMI 18.5–24.9 kg/m^2^	615/952 (64.6%)	628/947 (66.3%)	1243/1899 (65.5%)
BMI 25.0–29.9 kg/m^2^	230/952 (24.2%)	205/947 (21.6%)	435/1899 (22.9%)
BMI 30.0–40.0 kg/m^2^	107/952 (11.2%)	114/947 (12.0%)	221/1899 (11.6%)
GWG, kg ^a^	14.0 ± 5.3	14.0 ± 5.2	14.0 ± 5.3
Excessive GWG, *n* (%) ^b^	423/948 (44.6%)	422/940 (44.9%)	845/1888 (44.8%)
**Educational level, *n* (%)** ^c^			
General secondary school	131/951 (13.8%)	152/946 (16.1%)	283/1897 (14.9%)
Intermediate secondary school	412/951 (43.3%)	389/946 (41.1%)	801/1897 (42.2%)
High school	408/951 (42.9%)	405/946 (42.8%)	813/1897 (42.9%)
**Country of birth, *n* (%)**			
Germany	923/952 (97.0%)	914/945 (96.7%)	1837/1897 (96.8%)
Others	29/952 (3.0%)	31/945 (3.3%)	60/1897 (3.2%)
Caesarean section, *n* (%)	275/951 (28.9%)	256/946 (27.1%)	531/1897 (28.0%)
Preterm birth, *n* (%)	62/949 (6.5%)	55/947 (5.8%)	117/1896 (6.2%)
**Parity**			
Primiparous, *n* (%)	603/952 (63.3%)	509/947 (53.7%)	1112/1899 (58.6%)
Multiparous, *n* (%)			
1 more child	291/952 (30.6%)	346/947 (36.5%)	637/1899 (33.5%)
≥2 more children	58/952 (6.1%)	92/947 (9.7%)	150/1899 (7.9%)

Abbreviations: BMI: Body mass index; GWG: Gestational weight gain; SD: Standard deviation. ^a^ Mean ± SD (all such values). ^b^ Excessive GWG as classified by the IOM criteria [5]. ^c^ General secondary school: General school, which is completed through year 9; Intermediate secondary school: Vocational secondary school, which is completed through year 10; High school: Academic high school, which is completed through year 12 or 13.

**Table 2 nutrients-13-01310-t002:** Mean daily intake of selected food groups in the intervention and control groups.

	Time Point	Intervention Group		Control Group		*n* ^a^	Adjusted Effect Size ^b^ (95% CI)	Adjusted *p* Value ^b^
		***n***	**Mean ± SD**	***n***	**Mean ± SD**			
Caffeinated beverages (ml/day)	T1pp	897	172.17 ± 197.55	893	191.48 ± 235.54	1673	3.17 (−23.86, 30.20)	0.818
	T2pp	791	283.26 ± 339.91	776	286.97 ± 296.64	1483	10.10 (−4.74, 24.94)	0.182
	Time effect		*p* < 0.001 ^c^		*p* < 0.001 ^c^			
Soft drinks (ml/day)	T1pp	897	161.49 ± 301.67	893	246.42 ± 617.64	1674	−72.44 (−107.00, −37.88)	<0.001
	T2pp	791	160.57 ± 458.54	776	222.57 ± 592.33	1484	−41.04 (−90.10, 8.02)	0.101
	Time effect		*p* = 0.980 ^c^		*p* = 0.124 ^c^			
Vegetables (g/day)	T1pp	897	171.49 ± 156.65	891	157.74 ± 143.82	1670	12.81 (−4.21, 29.83)	0.140
	T2pp	790	200.39 ± 173.66	776	180.96 ± 150.26	1481	17.90 (3.53, 32.27)	0.015
	Time effect		*p* < 0.001 ^c^		*p* < 0.001 ^c^			
Fruit (g/day)	T1pp	897	206.62 ± 239.97	890	197.06 ± 194.46	1669	9.31 (−11.90, 30.51)	0.390
	T2pp	790	205.80 ± 192.72	776	194.52 ± 179.86	1481	8.62 (−14.11, 31.36)	0.457
	Time effect		*p* = 0.828 ^c^		*p* = 0.965 ^c^			
Nuts (g/day)	T1pp	891	3.13 ± 7.65	887	3.23 ± 13.69	1659	−0.12 (−0.50, 0.26)	0.537
	T2pp	791	2.18 ± 5.82	773	2.03 ± 6.02	1476	0.02 (−0.33, 0.37)	0.916
	Time effect		*p* < 0.001 ^c^		*p* = 0.009 ^c^			
Dairy products (g/day)	T1pp	898	344.12 ± 308.86	893	317.24 ± 302.89	1675	30.95 (6.70, 55.20)	0.012
	T2pp	791	298.96 ± 260.15	777	304.01 ± 327.06	1484	2.88 (−18.56, 24.32)	0.792
	Time effect		*p* < 0.001 ^c^		*p* = 0.172 ^c^			
Cheese (g/day)	T1pp	890	110.56 ± 116.24	883	107.21 ± 103.85	1652	1.39 (−13.90, 16.68)	0.859
	T2pp	788	97.15 ± 93.44	774	95.78 ± 103.77	1473	2.53 (−4.65, 9.72)	0.489
	Time effect		*p* = 0.003 ^c^		*p* = 0.005 ^c^			
Fish (g/day)	T1pp	897	17.02 ± 17.09	891	15.43 ± 19.01	1672	1.60 (0.81, 2.39)	<0.001
	T2pp	791	17.01 ± 16.53	775	16.46 ± 16.09	1483	0.10 (−0.59, 0.80)	0.768
	Time effect		*p* = 0.971 ^c^		*p* = 0.169 ^c^			
Meat and meat products (g/day)	T1pp	898	99.07 ± 63.20	893	100.64 ± 67.05	1675	−1.72 (−4.18, 0.73)	0.169
	T2pp	791	97.19 ± 55.68	777	95.53 ± 62.00	1484	1.17 (−5.67, 8.01)	0.738
	Time effect		*p* = 0.371 ^c^		*p* = 0.040 ^c^			
Sweets and snacks (g/day)	T1pp	898	96.34 ± 88.48	893	96.57 ± 115.94	1676	−0.04 (−10.83, 10.75)	0.994
	T2pp	791	78.22 ± 68.28	777	76.62 ± 58.13	1485	3.66 (−1.58, 8.90)	0.171
	Time effect		*p* < 0.001 ^c^		*p* < 0.001 ^c^			
Fast food (g/day)	T1pp	898	42.83 ± 33.80	893	43.77 ± 35.24	1675	−2.25 (−4.08, −0.42)	0.016
	T2pp	791	38.29 ± 26.73	777	43.17 ± 36.06	1484	−4.09 (−5.36, −2.82)	<0.001
	Time effect		*p* < 0.001 ^c^		*p* = 0.738 ^c^			

Abbreviations: T1pp: Assessment 6–8 weeks postpartum; T2pp: Assessment one year postpartum; SD: Standard deviation; CI: Confidence interval. ^a^ The total of participant numbers varies due to the applied covariates. ^b^ Linear regression models fit using generalised estimating equations adjusted for pre-pregnancy BMI category, age, parity, baseline assessment and time interval between questionnaire completion date and birth date of the child. ^c^ Linear mixed models for repeated measures adjusted for pre-pregnancy Body Mass Index (BMI) category, age and parity.

**Table 3 nutrients-13-01310-t003:** Food preparation and dietary choices in the intervention and control groups.

	Time Point	Intervention Group		Control Group		*n* ^a^	Adjusted OR ^b^ (95% CI)	Adjusted *p* Value ^b^
		*n*	%	*n*	%			
Whole grain bread	T1pp	828/893	92.7%	808/888	91.0%	1657	1.21 (0.65, 2.28)	0.546
	T2pp	745/788	94.5%	724/774	93.5%	1471	1.46 (1.07, 2.01)	0.017
	Time effect	*p* = 0.074 ^c^		*p* = 0.026 ^c^				
Rapeseed oil and olive oil (for meat and fish)	T1pp	456/729	62.6%	446/762	58.5%	1265	1.45 (1.13, 1.87)	0.004
	T2pp	397/615	64.6%	375/631	59.4%	1088	1.19 (1.04, 1.36)	0.011
	Time effect	*p* = 0.535 ^c^		*p* = 0.140 ^c^				
Rapeseed oil and olive oil (for vegetables)	T1pp	496/748	66.3%	477/766	62.3%	1259	1.32 (1.06, 1.64)	0.012
	T2pp	447/653	68.5%	408/653	62.5%	1107	1.32 (0.93, 1.88)	0.122
	Time effect	*p* = 0.215 ^c^		*p* = 0.556 ^c^				
Cooking at least 5 times per week	T1pp	536/893	60.0%	548/888	61.7%	1661	1.01 (0.87, 1.17)	0.914
	T2pp	578/771	75.0%	551/764	72.1%	1449	1.22 (0.97, 1.55)	0.095
	Time effect	*p* < 0.001 ^c^		*p* < 0.001 ^c^				
Vegetarian	T1pp	54/882	6.1%	47/886	5.3%	1677	1.20 (0.65, 2.23)	0.554
	T2pp	44/789	5.6%	36/774	4.7%	1513	1.30 (0.67, 2.55)	0.440
	Time effect	*p* = 0.717 ^c^		*p* = 0.446 ^c^				

Abbreviations: T1pp: Assessment 6–8 weeks postpartum; T2pp: Assessment one year postpartum; OR: Odds ratio. ^a^ The total of participant numbers varies due to the applied covariates. ^b^ Binary logistic regression models fit using generalised estimating equations adjusted for pre-pregnancy BMI category, age, parity, baseline assessment and time interval between questionnaire completion date and birth date of the child. ^c^ Linear mixed models for repeated measures adjusted for pre-pregnancy BMI category, age and parity.

**Table 4 nutrients-13-01310-t004:** Mean energy and macronutrient intake in the intervention and control groups.

	Time Point	Intervention Group		Control Group		*n* ^a^	Adjusted Effect Size ^b^ (95% CI)	Adjusted *p* Value ^b^
		***n***	**Mean ± SD**	***n***	**Mean ± SD**			
Energy (kcal/day)	T1pp	838	2236.99 ± 748.68	824	2227.84 ± 717.31	1440	1.55 (−93.15, 96.26)	0.974
	T2pp	736	2084.14 ± 645.78	706	2083.67 ± 693.66	1273	17.90 (−28.64, 64.43)	0.451
	Time effect		*p* < 0.001 ^c^		*p* < 0.001 ^c^			
Carbohydrates (E%)	T1pp	838	46.18 ± 8.50	824	46.77 ± 9.30	1440	−0.26 (−1.60, 1.07)	0.701
	T2pp	736	45.62 ± 9.32	706	46.32 ± 9.34	1273	−0.80 (−1.91, 0.32)	0.162
	Time effect		*p* = 0.084 ^c^		*p* = 0.424 ^c^			
Saccharose (g/day)	T1pp	838	53.85 ± 28.91	824	54.34 ± 30.53	1440	1.83 (0.89, 2.76)	<0.001
	T2pp	736	45.15 ± 24.43	706	46.12 ± 25.74	1273	−0.15 (−2.06, 1.77)	0.882
	Time effect		*p* < 0.001 ^c^		*p* < 0.001 ^c^			
Fibre (g/day)	T1pp	838	21.78 ± 9.51	824	20.98 ± 9.09	1440	0.34 (−0.62, 1.29)	0.491
	T2pp	736	21.63 ± 8.95	706	20.86 ± 8.93	1273	0.55 (−0.24, 1.35)	0.172
	Time effect		*p* < 0.001 ^c^		*p* < 0.001 ^c^			
Fat (E%)	T1pp	838	35.04 ± 7.49	824	34.55 ± 7.80	1440	0.23 (−0.93, 1.39)	0.698
	T2pp	736	34.77 ± 8.09	706	34.35 ± 7.66	1273	0.38 (−0.52, 1.27)	0.411
	Time effect		*p* = 0.230 ^c^		*p* = 0.096 ^c^			
Protein (E%)	T1pp	838	18.57 ± 4.07	824	18.40 ± 4.51	1440	0.17 (−0.13, 0.47)	0.262
	T2pp	736	18.86 ± 4.25	706	18.48 ± 4.23	1273	0.46 (0.05, 0.86)	0.028
	Time effect		*p* = 0.017 ^c^		*p* = 0.253 ^c^			
Alcohol (g/day)	T1pp	838	0.55 ± 1.57	824	0.79 ± 2.72	1440	−0.24 (−0.58, 0.10)	0.159
	T2pp	736	2.03 ± 2.59	706	2.34 ± 3.62	1273	−0.27 (−0.49, −0.06)	0.014
	Time effect		*p* < 0.001 ^c^		*p* < 0.001 ^c^			
HEI	T1pp	898	55.82 ± 8.37	893	54.81 ± 8.38	1676	0.64 (−0.11, 1.39)	0.093
	T2pp	791	56.22 ± 8.78	777	55.11 ± 8.33	1485	0.85 (0.03, 1.68)	0.043
	Time effect		*p* < 0.001 ^c^		*p* < 0.001 ^c^			

Abbreviations: HEI: Healthy Eating Index; T1pp: Assessment 6–8 weeks postpartum; T2pp: Assessment one year postpartum; SD: Standard deviation; CI: confidence interval. ^a^ The total of participant numbers varies due to the applied covariates. ^b^ Linear regression models fit using generalised estimating equations adjusted for pre-pregnancy BMI category, age, parity, baseline intake and time interval between questionnaire completion date and birth date of the child. ^c^ Linear mixed models for repeated measures adjusted for pre-pregnancy BMI category, age and parity.

**Table 5 nutrients-13-01310-t005:** Postpartum physical activity behaviour in the intervention and control groups.

	Time Point	Intervention Group		Control Group		*n* ^a^	Adjusted Effect Size ^b^ (95% CI)	Adjusted *p* Value ^b^
		*n*	Mean ± SD	*n*	Mean ± SD			
Total PA (MET-h/week)								
Total PA	T1pp	853	179.0 ± 63.9	863	183.0 ± 68.8	1538	−3.59 (−6.69, −0.50)	0.023
	T2pp	724	189.5 ± 64.7	737	188.1 ± 70.9	1338	2.60 (−5.23, 10.43)	0.515
	Time effect		*p* < 0.001 ^c^		*p* = 0.061 ^c^			
Total PA of light intensity and above	T1pp	856	168.9 ± 62.2	866	172.3 ± 67.5	1552	−2.58 (−5.56, 0.40)	0.090
	T2pp	725	182.2 ± 63.8	740	180.6 ± 70.1	1348	2.71 (−4.95, 10.37)	0.488
	Time effect		*p* < 0.001 ^c^		*p* < 0.001 ^c^			
Intensity (MET-h/week)								
Sedentary	T1pp	903	10.1 ± 10.0	899	10.7 ± 10.3	1664	−0.90 (−1.83, 0.03)	0.058
	T2pp	773	7.1 ± 5.9	759	7.6 ± 6.6	1446	−0.49 (−1.02, 0.05)	0.077
	Time effect		*p* < 0.001 ^c^		*p* < 0.001 ^c^			
Light-intensity	T1pp	890	87.6 ± 33.4	885	90.8 ± 33.7	1628	−1.37 (−5.08, 2.34)	0.469
	T2pp	753	105.4 ± 34.4	755	103.5 ± 34.4	1407	1.90 (−1.17, 4.97)	0.224
	Time effect		*p* < 0.001 ^c^		*p* < 0.001 ^c^			
Moderate- intensity	T1pp	867	80.5 ± 41.1	881	80.4 ± 43.5	1588	0.84 (−0.23, 1.92)	0.124
	T2pp	746	74.9 ± 43.0	746	75.1 ± 47.9	1387	1.00 (−3.35, 5.36)	0.651
	Time effect		*p* < 0.001 ^c^		*p* = 0.001 ^c^			
Vigorous- intensity	T1pp	901	1.2 ± 3.3	902	1.0 ± 3.3	1671	0.20 (−0.09, 0.50)	0.172
	T2pp	774	2.3 ± 4.7	766	2.0 ± 4.6	1456	0.32 (−0.14, 0.78)	0.170
	Time effect		*p* < 0.001 ^c^		*p* < 0.001 ^c^			
Type (MET-h/week)								
Household activity	T1pp	882	141.2 ± 55.2	887	145.1 ± 60.0	1625	−1.75 (−7.54, 4.04)	0.553
	T2pp	749	140.8 ± 54.2	751	137.6 ± 55.3	1402	4.13 (−2.23, 10.50)	0.203
	Time effect		*p* = 0.880 ^c^		*p* < 0.001 ^c^			
Occupational activity	T1pp	48	24.1 ± 38.9	52	18.6 ± 24.5	83	8.09 (1.51, 14.67)	0.016
	T2pp	218	42.1 ± 33.1	218	44.8 ± 36.2	365	−2.17 (−8.23, 3.88)	0.482
	Time effect		*p* = 0.002 ^c^		*p* < 0.001 ^c^			
Sport activity	T1pp	884	9.7 ± 8.2	888	9.2 ± 9.2	1627	0.12 (−0.55, 0.79)	0.725
	T2pp	756	12.5 ± 10.9	753	11.4 ± 10.5	1417	0.60 (−1.28, 2.47)	0.532
	Time effect		*p* < 0.001 ^c^		*p* < 0.001 ^c^			
Transportation activity	T1pp	902	14.3 ± 13.4	899	14.2 ± 13.4	1661	0.41 (−0.79, 1.61)	0.505
	T2pp	768	14.9 ± 12.9	762	15.2 ± 13.6	1439	0.01 (−1.09, 1.11)	0.986
	Time effect		*p* = 0.323 ^c^		*p* = 0.099 ^c^			
Inactivity	T1pp	900	12.7 ± 11.5	896	13.2 ± 11.3	1657	−0.75 (−1.85, 0.35)	0.182
	T2pp	771	9.6 ± 7.6	757	10.3 ± 8.3	1441	−0.63 (−1.35, 0.10)	0.091
	Time effect		*p* < 0.001 ^c^		*p* < 0.001 ^c^			
Meeting PA recommendations (*n* (%)) ^d^							OR (95% CI) ^e^	
	T1pp	466/884	52.7%	415/888	46.7%	1691	1.15 (0.99, 1.33)	0.060
	T2pp	439/756	58.1%	416/753	55.2%	1467	1.01 (0.83, 1.21)	0.957

Abbreviations: T1pp: Assessment 6–8 weeks postpartum; T2pp: Assessment one year postpartum; SD: Standard deviation; CI: Confidence interval; PA: Physical activity; OR: Odds ratio. ^a^ The total of participant numbers varies due to the applied covariates. ^b^ Linear regression models fit using generalised estimating equations adjusted for pre-pregnancy BMI category, age, parity, baseline assessment and time interval between questionnaire completion date and birth date of the child. ^c^ Linear mixed models for repeated measures adjusted for pre-pregnancy BMI category, age and parity. ^d^ Meeting recommendations defined as ≥ 7.5 MET-h/week in category sports activity of moderate-intensity or greater. ^e^ Binary logistic regression models fit using generalised estimating equations adjusted for pre-pregnancy BMI category, age, parity, baseline assessment and time interval between questionnaire completion date and birth date of the child.

## Data Availability

The datasets used and analysed during the current study are available from the corresponding author on reasonable request.

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
