# Peer review of "Effects of a Prenatal Lifestyle Intervention in Routine Care on Maternal Health Behaviour in the First Year Postpartum—Secondary Findings of the Cluster-Randomised GeliS Trial"

_nutrients, 2021, doi:10.3390/nu13041310_

Round 1
Reviewer 1 Report
I considered the manuscript very well written, the relevance of the study aim is suitably justified, the study design and statistical analyses is well done and results are clearly described. I also appreciated the sensitivity analyses presented in supplement tables.
However, I consider that you should discuss a be more the relevance of your results. For instance, the mean daily consumption of fish at T1pp is 17g in IG and 15,4g in CG. Considering the imprecision associated with a FFQ, do you consider this relevant? A similar comment could be done for the proportion of consumers of whole grain bread in T2pp or for HEI.
In table 5 you missed th units.
Reviewer 2 Report
This is a very well developed and written paper and study. The introduction provided an adequate description of previous studies and appropriate rationale. The methods, results and discussion were all written well and appropriately clear sections of the paper.
A few minor comments:
Is there a reason why breastfeeding wasn't included as a confounder? This would be reasonable given associations with post-partum health outcomes, although I understand not relevant to the actual intervention.
It would be helpful to have the sample size reported on earlier in the methods, perhaps even in the first paragraph of the methods.
Line 151/152; The question regarding smoking behaviour is unclear. Was this not asked to everyone in the study?
Lin2 161: It is unclear if baseline dietary and physical activity was included as an adjustment.
Figure 1: DO is an usual acronym. Suggest just spelling it out in full.
